# Security Performance Analysis of Downlink Double Intelligent Reflecting Surface Non-Orthogonal Multiple Access Network for Edge Users

**DOI:** 10.3390/s25041274

**Published:** 2025-02-19

**Authors:** Nguyen Thai Anh, Nguyen Hoang Viet, Dinh-Thuan Do, Adão Silva

**Affiliations:** 1Faculty of Electronics Technology, Industrial University of Ho Chi Minh City, Ho Chi Minh City 700000, Vietnam; vietnguyendhcn@gmail.com; 2Brenton School of Engineering, University of Mount Union, Alliance, OH 44601, USA; doth@mountunion.edu; 3Instituto de Telecomunicações (IT), University of Aveiro, 3810-193 Aveiro, Portugal; 4Departamento de Eletrónica, Telecomunicações e Informática (DETI), University of Aveiro, 3810-193 Aveiro, Portugal

**Keywords:** intelligent reflecting surface, non-orthogonal multiple access, physical layer security, secrecy outage probability, average secrecy rate

## Abstract

In this work, we study the security performance of a double intelligent reflecting surface non-orthogonal multiple access (DIRS-NOMA) wireless communication system supporting communication for a group of two NOMA users (UEs) at the edge, with the existence of an eavesdropping device (ED). We also assume that there is no direct connection between the BS and the UEs. From the proposed model, we compute closed-form expressions for the secrecy outage probability (SOP) and the average security rate (ASR) for each UE. After that, we discuss and analyze the system security performance according to the NOMA power allocation for each user and the number of IRS counter-emission elements. In addition, we analyze the SOP of both the considered DIRS-NOMA and conventional NOMA systems to demonstrate that DIRS-NOMA systems have much better security than conventional NOMA systems. Based on the analytical results, we develop an ASR optimization algorithm using the alternating optimization method, combining NOMA power allocation factor optimization and IRS passive beam optimization through the Lagrange double transform. The derived analytical expressions are validated through Monte Carlo simulations.

## 1. Introduction

It is known that an intelligent reflecting surface (IRS) is a simple way to manipulate the characteristics of electromagnetic waves (frequency, amplitude, and phase) and then reflect them to the desired destination. IRS technology certainly brings an additional level of coverage to wireless networks, and it has great potential to achieve the global connectivity required in future 6G systems [1]. The application of IRSs in wireless communication systems has been widely studied by the research community in recent years [2,3,4,5,6,7,8]. Nowadays, with the rapid development of wireless communication systems, requiring the connection of many devices and leading to limited wireless resources, a new technique is needed to meet user needs. Non-orthogonal multiple access (NOMA) is a promising technique that allows multiple users to share a single wireless resource. The advantage of NOMA is that it can serve additional secondary UEs through the specific channel that the primary UE is occupying. Although the performance of the primary UE may be affected by these secondary UEs, the overall throughput of the system can be significantly increased, especially if the secondary UE has a good connection to the BS. Unlike NOMA, cognitive radio (CR) relies on a communication technique in which the secondary UEs intelligently adjust their operating parameters to access the spectrum band occupied by the primary UE by cooperation or adaptation [9]. Integrated sensing and communications (ISAC) techniques can achieve energy savings, spectrum reuse, adaptability to changing conditions, and resilience to failures. In practice, NOMA, CR, and ISAC are all limited by inter-network interference and co-channel interference, which lead to serious degradation of reception reliability performance [10]. There are several security issues that need to be addressed in the implementation of NOMA, including the transmission power of the SIC implementation, the correct CSI, and the need for UE security and privacy. The relevant feature of NOMA here is that the power allocation coefficients are developed considering the channel characteristics of the legitimate UE, which ensures effective prevention of eavesdropping because SIC for eavesdroppers may not be feasible. Combining NOMA and IRS for wireless communication systems not only helps the wireless system achieve high connectivity but also has the ability to increase coverage. However, due to the broadcasting nature of physical devices, NOMA users are vulnerable to eavesdropping. In order to meet the security requirements of NOMA networks, existing security techniques such as artificial interference and cooperative relaying have been proposed to enhance system security, resist eavesdroppers, and increase system performance. Some of the pioneering research works on implementing IRS in NOMA networks to enhance the security of wireless systems include the following.

In [11], the security capacity of an IRS-enabled indoor wireless communication system was studied using an analytical approach that enables measurement of the ASR and SOP of the system considered. The closed-form analytical expressions for the SOP and ASR in the generalized tile-allocation-and-phase-shift-adjustment (TAaPSA) strategy were derived. Then, the optimal TAaPSA strategy, which aims to maximize ASR, was obtained using a genetic algorithm (GA). Furthermore, two realistic IRS cases, i.e., discrete phase-shift fixed amplitude (DPSFA) and discrete phase-dependent amplitude variation (DPDAV), were considered to investigate the performance loss caused by the IRS limitations. The simulation results confirmed the accuracy of the analytical results and the improvement of ASR using GA. The obtained simulation results show that the security performance was significantly improved with the help of an IRS and provide useful insights into the configuration of IRSs for high-security storage. In [12], the SOP of a single-antenna system enabled by RIS was studied in the presence of an eavesdropper. More specifically, an analytical expression of the SOP was derived and validated through simulation. The numerical results showed that the application of RIS can improve security performance. In [13], the security performance of an RIS-enabled communication system was studied in the presence of discrete phase changes. Using Fox’s H transforms, the authors obtained exact SOP and ASR expressions. The work concisely revealed the security diversity order and security array gain and quantified the security loss due to phase resolution.

In [14], the authors studied the SOP and security of a NOMA-RIS-enabled IoT system for a group of NOMA users in the presence of eavesdroppers. More specifically, the work presentd analytical expressions of SOP and security to provide secure analysis for two users. The authors applied a Golden Section search algorithm; the power allocation factors could be optimized at the base station to achieve the lowest SOP performance. The authors proposed a DNN-based prediction scheme that allows the source to adjust the power allocation factors to ensure fairness between two NOMA users as well as security. The numerical results showed that applying IRS with a higher metadata surface can significantly improve security performance. In particular, in [15], the authors presented an accurate approximation of the security ratio of IRS-enabled systems in the presence of discrete phase shifts and multiple eavesdroppers by exploiting the bounding technique in [16].

In [17], the authors formulated a transmission power minimization problem for a downlink DIRS-NOMA system according to the SINR of each UE, solved the non-convex optimization problem by establishing the relationship between the transmission power and the phase shift, and then developed an alternative optimization algorithm to determine the optimal phase shift. The results presented in the paper showed that the performance of dual IRS implementation is significantly better than that of centralized IRS implementation when there are enough reflectors. In [18], the authors studied a DIRS-NOMA network, in which they proposed a scheme to maximize the sum rate by simultaneously optimizing the time allocation and phase shift matrices. An alternative optimization algorithm was applied to increase the sum rate and efficiency of the dual IRS. In [19], the authors proposed a cooperative non-orthogonal multiple-input single-output access (MISO-NOMA) scheme for an ITS with an IRS and simultaneous wireless information and power transfer (SWIPT). In the paper, the UE fairness optimization problem was formulated to maximize the fairness ratio of UEs, depending on the service quality requirements of UEs with successive interference cancellation. The authors proposed an algorithm based on iterative successive convex approximation to simplify the non-convex objective function and then decomposed the original problem into two easily solved subproblems. Experimental results illustrated that the UE fairness of cooperative MISO-NOMA is better than that of both IRS-NOMA for ITS without SWIPT and IRS-OMA for ITS.

In [20], the performance of IRS-NOMA in mixed free space in the presence of imperfect CSI and imperfect SIC was studied, taking into account the harmful effects of atmospheric turbulence on the FSO communication link, affecting the end-to-end communication. The authors’ numerical analysis showed that the system performance could be significantly improved. In [21], the authors focused on the in-depth performance analysis of IRS-NOMA, multiplexing in the scope of SR systems, operating in k−μ fading channels. The authors formulated an exact expression of the OP related to the collective receiver framework. In addition, the authors presented an approximation technique for the OP using a univariate attenuation algorithm. The results were rigorously verified by the authors through Monte Carlo simulations. In [22], the authors proposed an AIRS-assisted unmanned aerial vehicle (UAV) relay scheme, in which the AIRS is equipped on the UAV to reflect signals from the ground base station (GBS) to the UE via NOMA access. The authors maximized the average total rate of the system by decomposing it into three subproblems through projection in the block coordinate system and proposed an iterative algorithm. In [23], the authors studied an active IRS-aided rate-splitting multiple access (RSMA). They grouped UEs for successive interference cancellation and proposed an alternating optimization (AO) algorithm with a novel AO strategy to solve the MMF OP optimization problem. The simulation results illustrated that the proposed scheme can bring energy efficiency to the system.

In [24], the physical layer security scheme of an IRS-assisted multiple-input single-output (MISO) wireless communication system was studied. The authors proposed the MISO-IRS security scheme, artificial noise (AN), and optimization problems. To solve the non-convex optimization problem, the authors transformed it into a two-layer optimization problem. The first layer consisted of finding the optimal slack variable, and the second layer consisted of finding the optimal beam shaping and phase shift for a certain value of the slack variable. The simulation results showed that the proposed scheme has good robustness and can significantly improve the security loss rate compared with the robust scheme without IRS or the robust scheme without optimal beam shaping. In [25], the authors studied the problem of deploying IRS to enhance the physical security layer in NOMA. The total security rate of NOMA-MIMO-IRS was maximized in the presence of eavesdroppers. The authors solved the convex optimization problem by using a rotation matrix and applied a particle swarm algorithm for global optimization. The simulation results verified that the proposed algorithm outperforms other methods. In [26], a MISO-IRS communication system that could simultaneously reflect and harvest energy from received signals was studied. The authors jointly designed the beam shaper at the access point (AP), the phase shift, and the energy harvesting schedule at the IRS to maximize the total rate of the system. The system performance analysis took into account the wireless energy harvesting capability of IRS elements, secure communication, and the impact of CSI. The simulation results of the paper showed that the total rate of the system affected the self-sustainability of the IRS, as did the number of energy-harvesting IRS elements.

### 1.1. Motivation and Contribution

Despite the rigorous efforts to date devoted to the performance of IRS-enhanced PLS, there are still very few studies on the security performance bounds in communication theory, due to the rather complex nature of the IRS aggregate fading model. In recent years, PLS has attracted widespread attention, and several techniques have been proposed to improve the security performance of wireless systems, such as cooperative diversity and spatial diversity. Since PLS takes advantage of the physical characteristics of the propagation medium, it is interesting to study the performance of PLS using IRS, while PLS for various wireless systems has been extensively studied in the literature [27,28], the application of IRS to improve PLS has not been studied in depth. To the best of our knowledge, the security performance for systems using IRS has only recently been considered in [29,30,31]. However, the authors focused their attention on design optimization, such as beam shaping and jamming. In this paper, we study the security performance of a DIRS-NOMA wireless communication system supporting communication with a group of two NOMA UEs at the edge, without direct connection with the BS and with one ED at each UE. The detailed contributions of this paper are as follows:We study a model of a DIRS-NOMA downlink wireless communication network system connecting a group of two UEs. The system consists of a BS supported by a DIRS connecting to a group of two NOMA UEs at the edge without a direct connection to the BS, using the NOMA technique. At each UE, there is an ED: ED1 is in the coverage area of IRS1, and IRS2 does not affect ED1; ED2 is in the coverage area of IRS2, and IRS1 does not affect ED2.Our analysis model considers statistical parameters of BS-DIRS-UE and BS-DIRS-ED channel models, taking into account the number of DIRS reflectors. It is assumed that BS-DIRS, DIRS-UE, and IRS-ED channels are interconnected channels with identical Rayleigh distributions.From the proposed model, we construct closed-form expressions of the SOP and ASR for each UE. We discuss and analyze the system security performance according to the NOMA power allocation for each UE and the number of IRS reflectors.Based on the analysis results, we build an ASR optimization algorithm using the alternating optimization method combining NOMA power allocation coefficient optimization and IRS passive beam optimization through the Lagrange double transform.Finally, we verify the inferred expressions by providing Monte Carlo simulations and analytical results.

### 1.2. Structure of the Paper

The paper is structured as follows. Section 2 presents the system model and problem formulation. Section 3 describes security performance analysis. Section 5 confirms the analytical results with simulation and discussions. Section 6 concludes the study.

The main notations of this paper are as follows: $\mathrm{Pr}\left(.\right)$ denotes the probability operator; $E\left[.\right]$ denotes the expectation operator; $f_{X}\left(.\right)$ and $F_{X}\left(.\right)$ denote the PDF and the CDF, respectively; $CN\left(.,.\right)$ is a circularly symmetric complex Gaussian distribution; ∼ stands for “distributed as”; $\left|.\right|$ is absolute operator; $\mathrm{Ei}\left(.\right)$ denotes the exponential integral function [32] [Equation (8.211.1)]. An identity matrix of size M×M is denoted by $I_{M}$; ${\left(.\right)}^{H}$ denotes the conjugate and Hermitian transpose; a diagonal matrix with $s_{1},\cdots ,s_{M}$ on the diagonal is denoted by $diag\left(s_{1},\cdots ,s_{M}\right)$. $\Gamma \left(\cdot \right)$ is the gamma function, and $\gamma \left(\cdot ,\cdot \right)$ is the lower incomplete gamma function. ${\left[y\right]}^{+}=\max\left\{0,y\right\}$.

## 2. System Model and Problem Formulation

### 2.1. The System Model

We study a downlink DIRS-NOMA network model for a two-UE NOMA edge group. The locations of the two-UE groups forming the NOMA user group in each cell are randomly selected among the UEs associated with that cell. One of these two UEs is considered UE1, and the other is considered UE2. For UE1, there is one ED1; in our scenario, ED1 is assumed to be supported only by IRS1. UE2 has one ED2, and ED2 is only supported by IRS2. In the downlink, in the BS-DIRS-UE link, it is assumed that all channels are flat-fading channels and that the BS has complete channel state information (CSI). We assume that IRS has no CSI about the ED link. Therefore, we consider a passive eavesdropping scenario. Consequently, we cannot guarantee secure data transmission. To quantify the performance, we use the SOP as the performance metric of interest. In addition, we assume that all links undergo independent Rayleigh fading. The BS-DIRS, DIRS-UE, and IRS-ED links can be LoS or NLoS for different scenarios. The channel gain between BS-DIRS is the same, denoted as $G\in C^{1\times N}$. The channel gain between DIRS-UE and IRS-ED is the same, denoted as $g\in C^{N\times 1}$. In particular, they are represented as the vector spaces $G=\left[G_{1},G_{2},\cdots ,G_{N}\right]$ and $g={\left[g_{1},g_{2},\cdots ,g_{N}\right]}^{T}$, respectively. All elements in *G* and *g* follow the fading Rayleigh model with variance $\sigma^{2}=1$. The BS is equipped with *M* antennas that generate K beamforming vectors to serve *K* groups of two NOMA UEs. The BS has one antenna supported by two IRSs to communicate with UE1 and UE2, each of which has a single antenna (Figure 1). Assume that there is no direct signal transmission between the BS and the UEs, due to the BS and the UEs being far apart or having obstacles between them. The IRS has *N* reflecting elements, and its reflection coefficient matrix is theoretically defined as $\beta_{i}\Theta =\beta_{i}diag\left(\theta_{1},\theta_{2},\cdots ,\theta_{N}\right)$, $F=\left\{\theta_{i}\left|\theta_{i}=e^{j\varphi_{i}},\varphi_{i}\in \left[0,2\pi \right)\right.\right\}$, in which $\beta_{i}\in \left[0,1\right]$ is called the amplitude of the reflection coefficient and $\theta_{i}\in [0,2\pi )$ is the phase shift angle of the kth adjustable IRS element (i=1,2,⋯,N).

### 2.2. Signal Analysis

The BS’s transmission signal is represented as $x=\sqrt{\alpha_{1}P_{s}}s_{UE1}+\sqrt{\alpha_{2}P_{s}}s_{UE2}$, where $P_{s}$ is the transmission power at the BS. $s_{UE1}$ and $s_{UE2}$ are the signals that the BS transmits to UE1 and UE2, respectively. $\alpha_{1}$ and $\alpha_{2}$ are the NOMA power allocation coefficient of the BS to UE1 and UE2, respectively. Parameters $\alpha_{1}$ and $\alpha_{2}$ satisfy the condition $\alpha_{1}+\alpha_{2}=1$. UEs are sorted based on their channel quality relative to the BS. According to the decoding principle of NOMA, UE1 performs SIC and directly decodes its own signal by treating UE2’s signal as interference. For UE2, it performs SIC and then decodes UE1’s signal and deletes this message from the observation area before decoding its own message [33,34]. Then, the signal received by the UEs and EDs can be represented as follows:The signal received by UE1 is(1)$$y_{UE1}=G\Theta g\left(d_{BR1}^{-\frac{\alpha_{G}}{2}}d_{11}^{-\frac{\alpha_{g}}{2}}+d_{BR2}^{-\frac{\alpha_{G}}{2}}d_{21}^{-\frac{\alpha_{g}}{2}}\right)x+n_{UE1};$$

The signal received by ED1 is


(2)
$$y_{ED1}=G\Theta gd_{BR1}^{-\frac{\alpha_{G}}{2}}d_{ED1}^{-\frac{\alpha_{g}}{2}}x+n_{ED1};$$


The signal received by UE2 is


(3)
$$y_{UE2}=G\Theta g\left(d_{BR2}^{-\frac{\alpha_{G}}{2}}d_{22}^{-\frac{\alpha_{g}}{2}}+d_{BR1}^{-\frac{\alpha_{G}}{2}}d_{12}^{-\frac{\alpha_{g}}{2}}\right)x+n_{UE2};$$


The signal received by ED2 is


(4)
$$y_{ED2}=G\Theta gd_{BR2}^{-\frac{\alpha_{G}}{2}}d_{ED2}^{-\frac{\alpha_{g}}{2}}x+n_{ED2}.$$


In these equations, $d_{BR1}$ is the signal transmission distance from BS to IRS1, $d_{BR2}$ is the signal transmission distance from BS to IRS2, $d_{11}$ is the distance between IRS1 and UE1, $d_{12}$ is the distance between IRS1 and UE2, $d_{21}$ is the distance between IRS2 and UE1, $d_{22}$ is the distance between IRS2 and UE2, $d_{ED1}$ is the distance from IRS1 to ED1, and $d_{ED2}$ is the distance from IRS2 to ED2. $n_{UE1}$, $n_{UE2}$, $n_{ED1}$, and $n_{ED2}$ are defined as the Gaussian noise at UE1, UE2, ED1, and ED2, respectively. $n_{UE1}$, $n_{UE2}$, $n_{ED1}$, and $n_{ED2}$ have the same variance of 1.

### 2.3. Signal-to-Noise Ratio Plus Noise for UEs and EDs

The signal-to-interference-plus-noise ratio (SINR) for the UEs and EDs is defined as follows:The SINR for UE1 is(5)$$SINR_{UE1}=\frac{{\left|G\Theta g\right|}^{2}R_{UE1}\alpha_{1}}{{\left|G\Theta g\right|}^{2}R_{UE1}\alpha_{2}+\frac{1}{\rho }};$$

The SINR for ED1 is


(6)
$$SINR_{ED1}=\frac{{\left|G\Theta g\right|}^{2}R_{ED1}\alpha_{1}}{{\left|G\Theta g\right|}^{2}R_{ED1}\alpha_{2}+\frac{1}{\rho }};$$


The SINR for UE2 is


(7)
$$SINR_{UE2}=\frac{{\left|G\Theta g\right|}^{2}R_{UE2}\alpha_{2}}{{\left|G\Theta g\right|}^{2}R_{UE2}\alpha_{1}+\frac{1}{\rho }};$$


The SINR at ED2 is


(8)
$$SINR_{ED2}=\frac{{\left|G\Theta g\right|}^{2}R_{ED2}\alpha_{2}}{{\left|G\Theta g\right|}^{2}R_{ED2}\alpha_{1}+\frac{1}{\rho }}.$$


In these equations, $R_{UE1}={\left(d_{BR1}^{-\frac{\alpha_{G}}{2}}d_{11}^{-\frac{\alpha_{g}}{2}}+d_{BR2}^{-\frac{\alpha_{G}}{2}}d_{21}^{-\frac{\alpha_{g}}{2}}\right)}^{2}$, $R_{ED1}={\left(d_{BR1}^{-\frac{\alpha_{G}}{2}}d_{ED1}^{-\frac{\alpha_{g}}{2}}\right)}^{2}$, $R_{UE2}=\left(d_{BR1}^{-\frac{\alpha_{G}}{2}}d_{12}^{-\frac{\alpha_{g}}{2}}+\right.$ ${\left.d_{BR2}^{-\frac{\alpha_{G}}{2}}d_{22}^{-\frac{\alpha_{g}}{2}}\right)}^{2}$, $R_{ED2}={\left(d_{BR2}^{-\frac{\alpha_{G}}{2}}d_{ED2}^{-\frac{\alpha_{g}}{2}}\right)}^{2}$, and $\rho =\frac{P_{s}}{\sigma_{UE}^{2}}$.

## 3. Security Performance Analysis

In this section, we analyze the security performance parameters of the DIRS-NOMA system, namely, the SOP and ASR.

### 3.1. Statistical Analysis

The statistical characteristics of BS-DIRS-UE and BS-IRS-ED channels are that we analyze the statistics of the random variable ${\left|G\Theta g\right|}^{2}$. ${\left|G\Theta g\right|}^{2}$ depends on the reflection parameters of the reflecting elements of the IRS; ${\left|G\Theta g\right|}^{2}$ is represented as ${\left|G\Theta g\right|}^{2}=\left|\sum_{k=1}^{N}\beta_{k}G_{k}g_{k}e^{j\theta_{k}}\right|$, where $G_{k}$ and $g_{k}$ are the channel gains of the kth element of *G* and *g*, respectively. Setting $\beta =\beta_{k}e^{j\theta_{k}}$, 0<β≤1, we have(9)$${\left|G\Theta g\right|}^{2}=\beta^{2}{\left(\sum_{k=1}^{N}\left|G_{k}\right|\left|g_{k}\right|\right)}^{2}.$$

Considering $X=\sum_{k=1}^{N}\left|G_{k}\right|\left|g_{k}\right|$, in which $\left|G_{k}\right|$ and $\left|g_{k}\right|$ are two random variables with Rayleigh distributions, it follows that the product $\left|G_{k}\right|\left|g_{k}\right|$ is also a random variable with a double Rayleigh distribution. Therefore, *X* is the sum of *N* random variables with statistically independent double Rayleigh distributions. The PDF and CDF of *X* can be approximated as follows [7]:(10)$$f_{X}\left(x\right)=\frac{x^{m}}{n^{m+1}\Gamma \left(m+1\right)}exp\left(-\frac{x}{n}\right),$$(11)$$F_{X}\left(x\right)=\frac{\gamma \left(m+1,\frac{x}{n}\right)}{\Gamma \left(m+1\right)}.$$
where $m=\frac{\Omega_{1}^{2}}{\Omega_{2}}-1$, $n=\frac{\Omega_{2}}{\Omega_{1}}$, $\Omega_{1}=N\frac{\pi }{2}$, and $\Omega_{2}=4N\left(1-\frac{\pi^{2}}{16}\right)$.

**Theorem 1.** 
*$E\left[X^{2}\right]$ and $E\left[X^{4}\right]$ are defined as follows:*

(12)
$$E[X^{2}]=\frac{n^{2}\Gamma \left(m+3\right)}{\Gamma \left(m+1\right)};$$


(13)
$$E\left[X^{4}\right]=\frac{n^{4}\Gamma \left(m+5\right)}{\Gamma \left(m+1\right)}.$$



**Proof.** Please refer to Appendix A.    □

### 3.2. SOP Analysis

The SOP is defined as the probability that a secure communication can be performed, which is a typical performance measure for PLS. Specifically, the instantaneous security rate can be expressed as(14)$$C_{UE}=\max\left\{\log_{2}\left(\frac{1+SINR_{UE}}{1+SINR_{ED}}\right),0\right\}.$$

UE1 SOP:


(15)
$$SOP_{UE1}=\mathrm{Pr}\left[C_{UE1}<C_{th1}\right].$$


The SOP of UE1 is presented in the following theorem.

**Theorem 2.** 
*The SOP of UE1 is defined as*

(16)
$$\begin{array}{cc} SOP_{UE1}=1 & -\frac{\gamma \left(m+1,\frac{1}{n}\sqrt{\frac{-B_{1}-\sqrt{\Delta_{1}}}{2A_{1}}}\right)}{\Gamma \left(m+1\right)}\\ \; & +\frac{\gamma \left(m+1,\frac{1}{n}\sqrt{\frac{-B_{1}+\sqrt{\Delta_{1}}}{2A_{1}}}\right)}{\Gamma \left(m+1\right)}, \end{array}$$

*where*

*$A_{1}=\left(1-2^{C_{th1}}\right)\rho^{2}\beta^{4}R_{UE1}R_{ED1}\alpha_{2}$,*

*$B_{1}=\rho \beta^{2}\left(\left(1-2^{C_{th1}}\alpha_{2}\right)R_{UE1}+\left(\alpha_{2}-2^{C_{th1}}\right)R_{ED1}\right)$,*

*$C_{1}=1-2^{C_{th1}}$,*

*$\Delta_{1}={\left(B_{1}\right)}^{2}-4A_{1}C_{1}$.*


**Proof.** Please refer toAppendix B.    □

UE2 SOP:


(17)
$$SOP_{UE2}=\mathrm{Pr}\left[C_{UE2}<C_{th2}\right].$$


The SOP of UE2 is presented in the following theorem.

**Theorem 3.** 
*The SOP of UE2 is defined as*

(18)
$$\begin{array}{cc} SOP_{UE2}=1 & -\frac{\gamma \left(m+1,\frac{1}{n}\sqrt{\frac{-B_{2}-\sqrt{\Delta_{2}}}{2A_{2}}}\right)}{\Gamma \left(m+1\right)}\\ \; & +\frac{\gamma \left(m+1,\frac{1}{n}\sqrt{\frac{-B_{2}+\sqrt{\Delta_{2}}}{2A_{2}}}\right)}{\Gamma \left(m+1\right)}, \end{array}$$

*where*

*$A_{2}=\left(1-2^{C_{th2}}\right)\rho^{2}\beta^{4}R_{UE2}R_{ED2}\alpha_{1}$,*

*$B_{2}=\left(1-2^{C_{th2}}\alpha_{1}\right)\rho \beta^{2}R_{UE2}+\left(\alpha_{1}-2^{C_{th2}}\right)\rho \beta^{2}R_{ED2}$,*

*$C_{2}=1-2^{C_{th2}}$,*

*$\Delta_{2}={\left(B_{2}\right)}^{2}-4A_{2}C_{2}$.*


**Proof.** Please refer to Appendix C.    □

### 3.3. Average Secrecy Rate (ASR)

The ASR is one of the most important performance metrics to evaluate a system’s covert performance in the presence of an eavesdropper. We assume that the worst-case eavesdropper can continue to achieve the largest channel gain with the maximum achievable ergodic rate $E\left[C_{ED1}\right]$. Therefore, the average achievable covert rate is [31].

UE1 ASR:


(19)
$$ASR_{UE1}={\left[E\left[C_{UE1}\right]-E\left[C_{ED1}\right]\right]}^{+}.$$


To determine $ASR_{UE1}$, we determine $E\left[C_{UE1}\right]$ and $E\left[C_{ED1}\right]$ as follows:(20)$$E\left[C_{UE1}\right]=E\left[\log_{2}\left(1+\frac{\beta^{2}R_{UE1}\alpha_{1}X^{2}}{\beta^{2}R_{UE1}\alpha_{2}X^{2}+\frac{1}{\rho }}\right)\right];$$(21)$$E\left[C_{ED1}\right]=E\left[\log_{2}\left(1+\frac{\beta^{2}R_{ED1}\alpha_{1}X^{2}}{\beta^{2}R_{ED1}\alpha_{2}X^{2}+\frac{1}{\rho }}\right)\right].$$

Transforming $E\left[C_{UE1}\right]$ and $E\left[C_{ED1}\right]$, we obtain(22)$$\begin{array}{cc} E\left[C_{UE1}\right] & =E\left[\log_{2}\left(\rho \beta^{2}X^{2}R_{UE1}+1\right)\right]\\ \; & -E\left[\log_{2}\left(\rho \beta^{2}X^{2}R_{UE1}\alpha_{2}+1\right)\right]\\ \; & =C_{UE1}^{1}-C_{UE1}^{2}; \end{array}$$(23)$$\begin{array}{cc} E\left[C_{ED1}\right] & =E\left[\log_{2}\left(\rho \beta^{2}X^{2}R_{ED1}+1\right)\right]\\ \; & -E\left[\log_{2}\left(\rho \beta^{2}X^{2}R_{ED1}\alpha_{2}+1\right)\right]\\ \; & =C_{ED1}^{1}-C_{ED1}^{2}. \end{array}$$

In these equations, $C_{UE1}^{1}$, $C_{UE1}^{2}$, $C_{ED1}^{1}$, and $C_{ED1}^{2}$ are determined by the following theorems.

**Theorem 4.** 
*$C_{UE1}^{1}$ and $C_{UE1}^{2}$ are determined by*

(24)
$$\begin{array}{cc} C_{UE1}^{1} & =\log_{2}\left(\rho \beta^{2}R_{UE1}\frac{\Gamma \left(m+3\right)n^{2}}{\Gamma \left(m+1\right)}+1\right)\\ \; & -\frac{{\left(\rho \beta^{2}R_{UE1}\right)}^{2}\left(\frac{\Gamma \left(m+5\right)n^{4}}{\Gamma \left(m+1\right)}-{\left(\frac{\Gamma \left(m+3\right)n^{2}}{\Gamma \left(m+1\right)}\right)}^{2}\right)}{2\ln2{\left(1+\rho \beta^{2}R_{UE1}\frac{\Gamma \left(m+3\right)n^{2}}{\Gamma \left(m+1\right)}\right)}^{2}}, \end{array}$$


(25)
$$\begin{array}{cc} C_{UE1}^{2} & =\log_{2}\left(\rho \beta^{2}R_{UE1}\alpha_{2}\frac{\Gamma \left(m+3\right)n^{2}}{\Gamma \left(m+1\right)}+1\right)\\ \; & -\frac{{\left(\rho \beta^{2}\alpha_{2}R_{UE1}\right)}^{2}\left(\frac{\Gamma \left(m+5\right)n^{4}}{\Gamma \left(m+1\right)}-{\left(\frac{\Gamma \left(m+3\right)n^{2}}{\Gamma \left(m+1\right)}\right)}^{2}\right)}{2\ln2{\left(1+\rho \beta^{2}\alpha_{2}R_{UE1}\frac{\Gamma \left(m+3\right)n^{2}}{\Gamma \left(m+1\right)}\right)}^{2}}. \end{array}$$



**Proof.** Please refer to Appendix D.    □

**Theorem 5.** 
*$C_{ED1}^{1}$ and $C_{ED1}^{2}$ are determined by*



(26)
$$\begin{array}{cc} C_{ED1}^{1} & =\log_{2}\left(\frac{\rho \beta^{2}R_{ED1}\Gamma \left(m+3\right)n^{2}}{\Gamma \left(m+1\right)}+1\right)\\ \; & -\frac{{\left(\rho \beta^{2}R_{ED1}\right)}^{2}\left(\frac{\Gamma \left(m+5\right)n^{4}}{\Gamma \left(m+1\right)}-{\left(\frac{\Gamma \left(m+3\right)n^{2}}{\Gamma \left(m+1\right)}\right)}^{2}\right)}{2\ln2{\left(1+\rho \beta^{2}R_{ED1}\frac{\Gamma \left(m+3\right)n^{2}}{\Gamma \left(m+1\right)}\right)}^{2}}; \end{array}$$



(27)
$$\begin{array}{cc} C_{ED1}^{2} & =\log_{2}\left(\frac{\rho \beta^{2}R_{ED1}\alpha_{2}\Gamma \left(m+3\right)n^{2}}{\Gamma \left(m+1\right)}+1\right)\\ \; & -\frac{{\left(\rho \beta^{2}\alpha_{2}R_{ED1}\right)}^{2}\left(\frac{\Gamma \left(m+5\right)n^{4}}{\Gamma \left(m+1\right)}-{\left(\frac{\Gamma \left(m+3\right)n^{2}}{\Gamma \left(m+1\right)}\right)}^{2}\right)}{2\ln2{\left(1+\frac{\rho \beta^{2}\alpha_{2}R_{ED1}\Gamma \left(m+3\right)n^{2}}{\Gamma \left(m+1\right)}\right)}^{2}}. \end{array}$$


**Proof.** Please refer to Appendix E.    □

UE2 ASR:


(28)
$$ASR_{UE2}={\left[E\left[C_{UE2}\right]-E\left[C_{ED2}\right]\right]}^{+}.$$


To determine $ASR_{UE2}$, we determine $E\left[C_{UE2}\right]$ and $E\left[C_{ED2}\right]$ as follows:(29)$$E\left[C_{UE2}\right]=E\left[\log_{2}\left(1+\frac{\beta^{2}R_{UE2}\alpha_{2}X^{2}}{\beta^{2}R_{UE2}\alpha_{1}X^{2}+\frac{1}{\rho }}\right)\right];$$(30)$$E\left[C_{ED2}\right]=E\left[\log_{2}\left(1+\frac{\beta^{2}R_{ED2}\alpha_{2}X^{2}}{\beta^{2}R_{ED1}\alpha_{1}X^{2}+\frac{1}{\rho }}\right)\right].$$
Transforming $E\left[C_{UE2}\right]$ and $E\left[C_{ED2}\right]$, we obtain(31)$$\begin{array}{cc} E\left[C_{UE2}\right] & =E\left[\log_{2}\left(\rho \beta^{2}X^{2}R_{UE2}+1\right)\right]\\ \; & -E\left[\log_{2}\left(\rho \beta^{2}X^{2}R_{UE2}\alpha_{1}+1\right)\right]\\ \; & =C_{UE2}^{1}-C_{UE2}^{2}; \end{array}$$(32)$$\begin{array}{cc} E\left[C_{ED2}\right] & =E\left[\log_{2}\left(\rho \beta^{2}X^{2}R_{ED2}+1\right)\right]\\ \; & -E\left[\log_{2}\left(\rho \beta^{2}X^{2}R_{ED2}\alpha_{1}+1\right)\right]\\ \; & =C_{ED2}^{1}-C_{ED2}^{2}. \end{array}$$
In these equations, $C_{UE2}^{1}$, $C_{UE2}^{2}$, $C_{ED2}^{1}$, and $C_{ED2}^{2}$ are determined by the following theorems.

**Theorem 6.** 
*$C_{UE2}^{1}$ and $C_{UE2}^{2}$ are determined by*

(33)
$$\begin{array}{cc} C_{UE2}^{1} & =\log_{2}\left(\frac{\rho \beta^{2}R_{UE2}\Gamma \left(m+3\right)n^{2}}{\Gamma \left(m+1\right)}+1\right)\\ \; & -\frac{{\left(\rho \beta^{2}R_{UE2}\right)}^{2}\left(\frac{\Gamma \left(m+5\right)n^{4}}{\Gamma \left(m+1\right)}-{\left(\frac{\Gamma \left(m+3\right)n^{2}}{\Gamma \left(m+1\right)}\right)}^{2}\right)}{2\ln2{\left(1+\rho \beta^{2}R_{UE2}\frac{\Gamma \left(m+3\right)n^{2}}{\Gamma \left(m+1\right)}\right)}^{2}}; \end{array}$$


(34)
$$\begin{array}{cc} C_{UE2}^{2} & =\log_{2}\left(\frac{\rho \beta^{2}R_{UE2}\alpha_{1}\Gamma \left(m+3\right)n^{2}}{\Gamma \left(m+1\right)}+1\right)\\ \; & -\frac{{\left(\rho \beta^{2}\alpha_{1}R_{UE2}\right)}^{2}\left(\frac{\Gamma \left(m+5\right)n^{4}}{\Gamma \left(m+1\right)}-{\left(\frac{\Gamma \left(m+3\right)n^{2}}{\Gamma \left(m+1\right)}\right)}^{2}\right)}{2\ln2{\left(1+\rho \beta^{2}\alpha_{1}R_{UE2}\frac{\Gamma \left(m+3\right)n^{2}}{\Gamma \left(m+1\right)}\right)}^{2}}. \end{array}$$



**Proof.** Please refer to Appendix F.    □

**Theorem 7.** 
*$C_{ED2}^{1}$ and $C_{ED2}^{2}$ are determined by*

(35)
$$\begin{array}{cc} C_{ED2}^{1} & =\log_{2}\left(\frac{\rho \beta^{2}R_{ED2}\Gamma \left(m+3\right)n^{2}}{\Gamma \left(m+1\right)}+1\right)\\ \; & -\frac{{\left(\rho \beta^{2}R_{ED2}\right)}^{2}\left(\frac{\Gamma \left(m+5\right)n^{4}}{\Gamma \left(m+1\right)}-{\left(\frac{\Gamma \left(m+3\right)n^{2}}{\Gamma \left(m+1\right)}\right)}^{2}\right)}{2\ln2{\left(1+\rho \beta^{2}R_{ED2}\frac{\Gamma \left(m+3\right)n^{2}}{\Gamma \left(m+1\right)}\right)}^{2}}; \end{array}$$


(36)
$$\begin{array}{cc} C_{ED2}^{2} & =\log_{2}\left(\frac{\rho \beta^{2}R_{ED2}\alpha_{1}\Gamma \left(m+3\right)n^{2}}{\Gamma \left(m+1\right)}+1\right)\\ \; & -\frac{{\left(\rho \beta^{2}\alpha_{1}R_{ED2}\right)}^{2}\left(\frac{\Gamma \left(m+5\right)n^{4}}{\Gamma \left(m+1\right)}-{\left(\frac{\Gamma \left(m+3\right)n^{2}}{\Gamma \left(m+1\right)}\right)}^{2}\right)}{2\ln2{\left(1+\rho \beta^{2}\alpha_{1}R_{ED2}\frac{\Gamma \left(m+3\right)n^{2}}{\Gamma \left(m+1\right)}\right)}^{2}}. \end{array}$$



**Proof.** Please refer to Appendix G.    □

## 4. Weighted Secrecy Rate Optimization Problem

In this subsection, we optimize the weighted secrecy rate (WSR) of a legitimate user subject to the total power constraint of the BS, the IRS phase-shift constraints, and the SOP/SIC constraints of the legitimate user by designing the NOMA transmission power allocation factor of the BS and the IRS’s reflected beamforming vector Θ. To enhance the security of the above system from the perspective of the physical layer, we jointly optimize $\alpha_{j}$ (j=1;2) and Θ so that the system’s secrecy rate is optimized. Specifically, secrecy rate optimization can be achieved by the optimization problem stated as follows [35,36,37,38]:(37a)$$\begin{array}{cc} \; & \mathrm{Maximize}ASR_{UE}\left(\alpha_{j},\Theta \right), \end{array}$$(37b)$$\begin{array}{cc} \; & \mathrm{subject}\;\mathrm{to}\sum_{j=1}^{2}\alpha_{j}=1;0\leq \alpha_{j}, \end{array}$$(37c)$$\begin{array}{cc} \; & \left|\theta_{i}\right|\in F,i=1,\cdots ,N. \end{array}$$

### 4.1. Weighted Secrecy Rate Optimization for UE1

$ASR_{UE1}\left(\alpha_{1},\Theta \right)$ is defined as(38)$$\begin{array}{c} ASR_{UE1}\left(\alpha_{1},\Theta \right)={\left[\log_{2}\left(\frac{1+\frac{{\left|G\Theta g\right|}^{2}R_{UE1}\alpha_{1}}{{\left|G\Theta g\right|}^{2}R_{UE1}\alpha_{2}+\frac{1}{\rho }}}{1+\frac{{\left|G\Theta g\right|}^{2}R_{ED1}\alpha_{1}}{{\left|G\Theta g\right|}^{2}R_{ED1}\alpha_{2}+\frac{1}{\rho }}}\right)\right]}^{+}. \end{array}$$
Transforming expression $ASR_{UE1}\left(\alpha_{1},\Theta \right)$, we obtain(39)$$ASR_{UE1}\left(\alpha_{1},\Theta \right)={\left[\log_{2}\left(1+\Upsilon_{UE1}\right)\right]}^{+},$$
where(40)$$\Upsilon_{UE1}=\frac{D_{H1}\alpha_{1}{\left|G\Theta g\right|}^{2}}{D_{T1}{\left|G\Theta g\right|}^{4}\alpha_{2}+D_{\alpha_{2}}{\left|G\Theta g\right|}^{2}+1}.$$
with $D_{H1}=\left(R_{UE1}-R_{ED1}\right)\rho$, $D_{T1}=\rho^{2}R_{UE1}R_{ED1}$, and $D_{\alpha_{2}}=\left(\alpha_{2}R_{UE1}+R_{ED1}\right)\rho$.

Our goal is to optimize the WSR of UE1 by combining the design of the transmision beamforming matrix $\alpha_{1}$ at the BS and Θ at the IRS. Therefore, the WSR optimization problem is formulated as follows:(41a)$$\begin{array}{cc} \; & \mathrm{Maximize}ASR_{UE1}\left(\alpha_{1},\Theta \right)=\omega_{UE1}\left[\log_{2}\left(1+\Upsilon_{UE1}\right)\right], \end{array}$$(41b)$$\begin{array}{cc} \; & \mathrm{subject}\;\mathrm{to}\sum_{j=1}^{2}\alpha_{j}=1;0\leq \alpha_{j}, \end{array}$$(41c)$$\begin{array}{cc} \; & \left|\theta_{i}\right|\in F,i=1,\cdots ,N \end{array}$$
where the weight $\omega_{UE1}$ is used to represent the priority of UE1.

The objective function $ASR_{UE1}\left(\alpha_{1},\Theta \right)$ is non-convex, because the constraint set *F* is non-convex. To solve the logarithm of the objective function (41a)–(41c), we apply the Lagrangian double transform proposed in [31]. Then, (41a)–(41c) can be equivalently written as follows:(42a)$$\begin{array}{cc} \; & \underset{\alpha_{1},\Theta ,\lambda_{UE1}}{\mathrm{Maximize}}ASR_{UE1}\left(\alpha_{1},\Theta ,\lambda_{UE1}\right), \end{array}$$(42b)$$\begin{array}{cc} \; & \mathrm{subject}\;\mathrm{to}\sum_{j=1}^{2}\alpha_{j}=1;0\leq \alpha_{j}, \end{array}$$(42c)$$\begin{array}{cc} \; & \left|\theta_{i}\right|\in F,i=1,\cdots ,N \end{array}$$
where $\lambda_{UE1}$ is an auxiliary variable for decoding $\Upsilon_{UE1}$; the new objective function is defined by(43)$$\begin{array}{cc} ASR_{UE1} & \left(\alpha_{1},\Theta ,\lambda_{UE1}\right)=\omega_{UE1}\log\left(1+\lambda_{UE1}\right)\\ \; & -\omega_{UE1}\lambda_{UE1}+\frac{\omega_{UE1}\left(1+\lambda_{UE1}\right)\Upsilon_{UE1}}{1+\Upsilon_{UE1}}. \end{array}$$
In (42), when $\alpha_{1}$, Θ are fixed, the optimal $\lambda_{UE1}$ is the solution of the equation(44)$$-\omega_{UE1}\lambda_{UE1}+\frac{\omega_{UE1}\left(1+\lambda_{UE1}\right)\Upsilon_{UE1}}{1+\Upsilon_{UE1}}=0.$$
We find the optimal $\lambda_{UE1}$ as follows:(45)$$\lambda_{UE1}^{*}=\Upsilon_{UE1}.$$

In the following, we solve $\alpha_{1}$ by fixing Θ and solve Θ by fixing $\alpha_{1}$, respectively. Then, the original problem (42a)–(42c) can be solved iteratively by applying the alternating optimization method as illustrated in Figure 2. Specifically, in each iteration, we first update the nominal $\Upsilon_{UE1}$; then, the better solutions for $\alpha_{1}$ and Θ are updated accordingly. This process is repeated until no further improvement is achieved.

#### 4.1.1. Optimizing NOMA Power Allocation Factor for UE1

We study how to find the optimization of $\alpha_{1}$ with fixed Θ for (42a)–(42c). Hence, with fixed $\lambda_{UE1}$ and Θ, the optimization of $\alpha_{1}$ becomes as follows:(46a)$$\begin{array}{cc} \; & \underset{\alpha_{1}}{\mathrm{Maximize}}f_{UE1}\left(\alpha_{1}\right), \end{array}$$(46b)$$\begin{array}{cc} \; & \mathrm{subject}\;\mathrm{to}\sum_{j=1}^{2}\alpha_{j}=1;0\leq \alpha_{j}, \end{array}$$
where $f_{UE1}\left(\alpha_{1}\right)=\frac{\omega_{UE1}\left(1+\lambda_{UE1}\right)\Upsilon_{UE1}}{1+\Upsilon_{UE1}}$.

Using the quadratic transformation proposed in [32], $f_{UE1}\left(\alpha_{1}\right)$ is re-expressed as(47)$$\begin{array}{cc} f_{UE1a}\left(\alpha_{1},\beta_{UE1}\right) & =2\beta_{UE1}\sqrt{\omega_{UE1}\left(1+\lambda_{UE1}\right)\Upsilon_{UE1}}\\ \; & -\beta_{UE1}^{2}\left(1+\Upsilon_{UE1}\right). \end{array}$$

The optimization problem in (46a) and (46b) is solved by updating $\alpha_{1}$ and $\beta_{UE1}$ while fixing one of them. The optimal $\beta_{UE1}$ is determined as(48)$$\beta_{UE1}^{o}=\frac{\sqrt{\omega_{UE1}\left(1+\lambda_{UE1}\right)\Upsilon_{UE1}}}{\left(1+\Upsilon_{UE1}\right)},$$
where $\beta_{UE1}^{o}$ is the solution of the equation $\frac{\partial f_{UE1a}\left(\alpha_{1},\beta_{UE1}\right)}{\partial \beta_{UE1}}=0$. Then, fixing $\beta_{UE1}$, the optimal $\alpha_{1}$ is determined by the following theorem.

**Theorem 8.** 
*The optimal $\alpha_{1}$ is determined by*

(49)
$$\alpha_{1}^{o}=\frac{{\left(\frac{\sqrt{\omega_{UE1}\left(1+\lambda_{UE1}\right)}}{\beta_{UE1}}\right)}^{2}m_{3}}{\left(m_{1}+m_{2}{\left(\frac{\sqrt{\omega_{UE1}\left(1+\lambda_{UE1}\right)}}{\beta_{UE1}}\right)}^{2}\right)},$$

*where $m_{1}=\left(R_{UE1}-R_{ED1}\right)\rho {\left|G\Theta g\right|}^{2}$, $m_{2}=\rho^{2}{\left|G\Theta g\right|}^{4}R_{UE1}R_{ED1}+R_{UE1}\rho {\left|G\Theta g\right|}^{2}$, and $m_{3}=\rho^{2}{\left|G\Theta g\right|}^{4}R_{UE1}R_{ED1}+R_{UE1}\rho {\left|G\Theta g\right|}^{2}+R_{ED1}\rho {\left|G\Theta g\right|}^{2}+1$.*


**Proof.** Please refer to Appendix H.    □

#### 4.1.2. Optimized Θ Reflection Matrix for UE1

We study how to find the optimization of $\alpha_{1}$ with fixed Θ for (42a)–(42c). Hence, with fixed $\lambda_{UE1}$ and Θ, the optimization of $\alpha_{1}$ becomes as follows:(50a)$$\begin{array}{cc} \; & \underset{\Theta }{\mathrm{Maximize}}f_{UE1}\left(\Theta \right), \end{array}$$(50b)$$\begin{array}{cc} \; & \left|\theta_{i}\right|\in F,i=1,\cdots ,N \end{array}$$
where $f_{UE1}\left(\Theta \right)=\frac{\omega_{UE1}\left(1+\lambda_{UE1}\right)\Upsilon_{UE1}}{1+\Upsilon_{UE1}}$.

We optimize Θ in (42a)–(42c) given fixed values of $\lambda_{UE1}$ and $\alpha_{1}$. Using $\Upsilon_{UE1}$ as defined in (40), the objective function of (50a) and (50b) is expressed as a function of Θ:(51)$$f_{UE1}\left({\left|G\Theta g\right|}^{2}\right)=\frac{\omega_{UE1}\left(1+\lambda_{UE1}\right)D_{H1}\alpha_{1}}{{\left|G\Theta g\right|}^{2}D_{T1}\alpha_{2}+\frac{1}{{\left|G\Theta g\right|}^{2}}+D_{\alpha_{2s}}},$$
where $D_{\alpha_{2s}}=\left(R_{UE1}+\alpha_{2}R_{ED1}\right)\rho$. Transforming expression (51), we obtain(52)$$f_{UE1}\left({\left|G\Theta g\right|}^{2}\right)=\frac{\omega_{UE1}\left(1+\lambda_{UE1}\right)D_{H1}\alpha_{1}}{{\left|G\Theta g\right|}^{2}D_{T1}\alpha_{2}+\frac{1}{{\left|G\Theta g\right|}^{2}}+D_{\alpha_{2s}}}.$$
The maximum value of $f_{UE1}\left({\left|G\Theta g\right|}^{2}\right)$ is determined by(53)$$\mathrm{Max}\left|f_{UE1}\left({\left|G\Theta g\right|}^{2}\right)\right|=\frac{\omega_{UE1}\left(1+\lambda_{UE1}\right)\alpha_{1}D_{H1}}{2\sqrt{D_{T1}\alpha_{2}}+D_{\alpha_{2s}}}$$
when ${\left|G\Theta g\right|}^{2}=\frac{1}{\sqrt{\rho^{2}R_{UE1}R_{ED1}\left(1-\alpha_{1}\right)}}$. Θ is optimally adjusted so that(54)$${\left|G\Theta g\right|}^{2}=\frac{1}{\sqrt{\rho^{2}R_{UE1}R_{ED1}\left(1-\alpha_{1}\right)}}.$$
An alternating optimization method to optimize $ASR_{UE1}$ for UE1 is proposed by us in Algorithm 1.
**Algorithm 1:** Alternating optimization method for solving (42a)–(42c).1: **Step 0:** Initialize $\alpha_{1}^{\left(0\right)}$ and $\Theta^{\left(0\right)}$ to feasible values.**Repeat**2: **Step 1:** Update the nominal $\lambda_{UE1}^{*}=\Upsilon_{UE1}^{\left(i\right)}$ by (45).3: **Step 2.1:** Update $\beta_{UE1}^{o\left(i\right)}$ by (48).4: **Step 2.2:** Update the transmission beamforming $\alpha_{1}^{o\left(i\right)}$ by (49).5: **Step 3:** Update $\Theta^{\left(i\right)}$ by (54).Until the value of the function $ASR_{UE1}\left(\alpha_{1},\Theta ,\lambda_{UE1}\right)$ in (43) converges.

### 4.2. Weighted Secrecy Rate Optimization for UE2

$ASR_{UE2}\left(\alpha_{2},\Theta \right)$ is defined as(55)$$ASR_{UE2}\left(\alpha_{2},\Theta \right)={\left[\log_{2}\left(\frac{1+\frac{{\left|G\Theta g\right|}^{2}R_{UE2}\alpha_{2}}{{\left|G\Theta g\right|}^{2}R_{UE2}\alpha_{1}+\frac{1}{\rho }}}{1+\frac{{\left|G\Theta g\right|}^{2}R_{ED2}\alpha_{2}}{{\left|G\Theta g\right|}^{2}R_{ED2}\alpha_{1}+\frac{1}{\rho }}}\right)\right]}^{+}.$$
Transforming expression $ASR_{UE2}\left(\alpha_{2},\Theta \right)$, we obtain(56)$$ASR_{UE2}\left(\alpha_{2},\Theta \right)={\left[\log_{2}\left(1+\Upsilon_{UE2}\right)\right]}^{+},$$
where(57)$$\Upsilon_{UE2}=\frac{\alpha_{2}D_{H2}{\left|G\Theta g\right|}^{2}}{{\left|G\Theta g\right|}^{4}D_{T2}\alpha_{1}+D_{\alpha_{1}}{\left|G\Theta g\right|}^{2}+1},$$
with $D_{H2}=\left(R_{UE2}-R_{ED2}\right)\rho$, $D_{T2}=\rho^{2}R_{UE2}R_{ED2}$, and $D_{\alpha_{1}}=\left(\alpha_{1}R_{UE2}+R_{ED2}\right)\rho$.

Our goal is to optimize the WSR of UE2 by combining the design of the transmission beamforming matrix $\alpha_{2}$ at the BS and Θ at the IRS. Therefore, the WSR optimization problem is formulated as(58a)$$\begin{array}{cc} \; & \mathrm{Maximize}ASR_{UE2}\left(\alpha_{2},\Theta \right)=\omega_{UE2}\left[\log_{2}\left(1+\Upsilon_{UE2}\right)\right], \end{array}$$(58b)$$\begin{array}{cc} \; & \mathrm{subject}\;\mathrm{to}\sum_{j=1}^{2}\alpha_{j}=1;0\leq \alpha_{j}, \end{array}$$(58c)$$\begin{array}{cc} \; & \left|\theta_{i}\right|\in F,i=1,\cdots ,N \end{array}$$
where the weight $\omega_{UE2}$ is used to represent the priority of UE2.

The objective function $ASR_{UE2}\left(\alpha_{2},\Theta \right)$ is non-convex, because the constraint set *F* is non-convex. To solve the logarithm of the objective function (58a)–(58c), we apply the Lagrangian double transform proposed in [32]. Then, (58a)–(58c) can be equivalently written as follows:(59a)$$\begin{array}{cc} \; & \underset{\alpha_{2},\Theta ,\lambda_{UE2}}{\mathrm{Maximize}}ASR_{UE2}\left(\alpha_{2},\Theta ,\lambda_{UE2}\right), \end{array}$$(59b)$$\begin{array}{cc} \; & \mathrm{subject}\;\mathrm{to}\sum_{j=1}^{2}\alpha_{j}=1;0\leq \alpha_{j}, \end{array}$$(59c)$$\begin{array}{cc} \; & \left|\theta_{i}\right|\in F,i=1,\cdots ,N \end{array}$$
where $\lambda_{UE2}$ is an auxiliary variable for decoding $\Upsilon_{UE2}$; the new objective function is defined by(60)$$\begin{array}{cc} ASR_{UE2} & \left(\alpha_{2},\Theta ,\lambda_{UE2}\right)=\omega_{UE2}\log\left(1+\lambda_{UE2}\right)\\ \; & -\omega_{UE2}\lambda_{UE2}+\frac{\omega_{UE2}\left(1+\lambda_{UE2}\right)\Upsilon_{UE2}}{1+\Upsilon_{UE2}}. \end{array}$$
In (59a)–(59c), when $\alpha_{2}$ and Θ are fixed, the optimal $\lambda_{UE2}$ is the solution of the equation:(61)$$-\omega_{UE2}\lambda_{UE2}+\frac{\omega_{UE2}\left(1+\lambda_{UE2}\right)\Upsilon_{UE2}}{1+\Upsilon_{UE2}}=0.$$
We find the optimal $\lambda_{UE2}$ as follows(62)$$\lambda_{UE2}^{*}=\Upsilon_{UE2}.$$

In the following, we solve $\alpha_{2}$ by fixing Θ and solve Θ by fixing $\alpha_{2}$, respectively. Then, the original problem (59a)–(59c) can be solved iteratively by applying the alternating optimization method as illustrated in Figure 3. Specifically, in each iteration, we first update the nominal $\Upsilon_{UE2}$, then the better solutions for $\alpha_{2}$ and Θ are updated accordingly. This process is repeated until no further improvement is achieved.

#### 4.2.1. Optimizing NOMA Power Allocation Factor for UE2

We study how to find the optimization of $\alpha_{2}$ with fixed Θ for (59a)–(59c). Hence, with fixed $\lambda_{UE2}$ and Θ, the optimization of $\alpha_{2}$ becomes(63a)$$\begin{array}{cc} \; & \underset{\alpha_{2}}{\mathrm{Maximize}}f_{UE2}\left(\alpha_{2}\right), \end{array}$$(63b)$$\begin{array}{cc} \; & \mathrm{subject}\mathrm{to}:\sum_{j=1}^{2}\alpha_{j}=1;0\leq \alpha_{j}. \end{array}$$
where $f_{UE2}\left(\alpha_{2}\right)=\frac{\omega_{UE2}\left(1+\lambda_{UE2}\right)\Upsilon_{UE2}}{1+\Upsilon_{UE2}}$.

Using the quadratic transformation proposed in [32], $f_{UE2}\left(\alpha_{2}\right)$ is re-expressed as(64)$$\begin{array}{cc} f_{UE2a}\left(\alpha_{2},\beta_{UE2}\right)= & 2\beta_{UE2}\sqrt{\omega_{UE2}\left(1+\lambda_{UE2}\right)\Upsilon_{UE2}}\\ \; & -\beta_{UE2}^{2}\left(1+\Upsilon_{UE2}\right). \end{array}$$

At this point, (63a) and (63b) is solved by updating $\alpha_{2}$ and $\beta_{UE2}$ by fixing one of them. The optimal $\beta_{UE2}$ is determined as follows:(65)$$\beta_{UE2}^{o}=\frac{\sqrt{\omega_{UE2}\left(1+\lambda_{UE2}\right)\Upsilon_{UE2}}}{\left(1+\Upsilon_{UE2}\right)}.$$
where $\beta_{UE2}^{o}$ is the solution of equation $\frac{\partial f_{UE2a}\left(\alpha_{2},\beta_{UE2}\right)}{\partial \beta_{UE2}}=0$. Then, fixing $\beta_{UE2}$, the optimal $\alpha_{2}$ is determined by the following theorem.

**Theorem 9.** 
*The optimal $\alpha_{2}$ is determined by*

(66)
$$\alpha_{2}^{o}=\frac{{\left(\frac{\sqrt{\omega_{UE2}\left(1+\lambda_{UE2}\right)}}{\beta_{UE2}}\right)}^{2}n_{3}}{\left(n_{1}+n_{2}{\left(\frac{\sqrt{\omega_{UE2}\left(1+\lambda_{UE2}\right)}}{\beta_{UE2}}\right)}^{2}\right)},$$

*where $n_{1}=\left(R_{UE2}-R_{ED2}\right)\rho {\left|G\Theta g\right|}^{2}$, $n_{2}=\rho^{2}{\left|G\Theta g\right|}^{4}R_{UE1}R_{ED2}+R_{UE2}\rho {\left|G\Theta g\right|}^{2}$, and $n_{3}=\rho^{2}{\left|G\Theta g\right|}^{4}R_{UE2}R_{ED2}+R_{UE2}\rho {\left|G\Theta g\right|}^{2}+R_{ED2}\rho {\left|G\Theta g\right|}^{2}+1$.*


**Proof.** Please refer to Appendix I.    □

#### 4.2.2. Optimized Θ Reflection Matrix for UE2

We study how to find the optimization of $\alpha_{2}$ with fixed Θ for (59a)–(59c). Hence, with fixed $\lambda_{UE2}$ and Θ, the optimization of $\alpha_{2}$ becomes the following:(67a)$$\begin{array}{cc} \; & \underset{\Theta }{\mathrm{Maximize}}f_{UE2}\left(\Theta \right), \end{array}$$(67b)$$\begin{array}{cc} \; & \left|\theta_{i}\right|\in F,i=1,\cdots ,N \end{array}$$
where $f_{UE2}\left(\Theta \right)=\frac{\omega_{UE2}\left(1+\lambda_{UE2}\right)\Upsilon_{UE2}}{1+\Upsilon_{UE2}}$.

We optimize Θ in (65) given a fixed $\lambda_{UE2}$ and $\alpha_{2}$. Using $\Upsilon_{UE2}$ defined in (57), the objective function of (67a) and (67b) is expressed as a function of Θ:(68)$$f_{UE2}\left({\left|G\Theta g\right|}^{2}\right)=\frac{\omega_{UE2}\left(1+\lambda_{UE2}\right)D_{H2}\alpha_{2}{\left|G\Theta g\right|}^{2}}{D_{T2}{\left|G\Theta g\right|}^{4}\alpha_{1}+D_{\alpha_{1s}}{\left|G\Theta g\right|}^{2}+1}.$$
Transforming expression (66), we obtain(69)$$f_{UE2}\left({\left|G\Theta g\right|}^{2}\right)=\frac{\omega_{UE2}\left(1+\lambda_{UE2}\right)\alpha_{2}D_{H2}}{{\left|G\Theta g\right|}^{2}D_{T2}\alpha_{1}+\frac{1}{{\left|G\Theta g\right|}^{2}}+D_{\alpha_{1s}}}.$$
The maximum value of $f_{UE1}\left({\left|G\Theta g\right|}^{2}\right)$ is determined by(70)$$\mathrm{Max}\left|f_{UE2}\left({\left|G\Theta g\right|}^{2}\right)\right|=\frac{\omega_{UE2}\left(1+\lambda_{UE2}\right)\alpha_{2}D_{H2}}{2\sqrt{D_{T2}\alpha_{1}}+D_{\alpha_{1s}}}$$
when ${\left|G\Theta g\right|}^{2}=\frac{1}{\sqrt{D_{T2}\left(1-\alpha_{2}\right)}}$. Θ is optimally adjusted so that(71)$${\left|G\Theta g\right|}^{2}=\frac{1}{\sqrt{\rho^{2}R_{UE2}R_{ED2}\left(1-\alpha_{2}\right)}}.$$
An alternating optimization method to optimize $ASR_{UE2}$ for UE2 is proposed by us in Algorithm 2.
**Algorithm 2:** Alternating optimization method for solving (61).1: **Step 0:** Initialize $\alpha_{2}^{\left(0\right)}$ and $\Theta^{\left(0\right)}$ to feasible values.**Repeat**2: **Step 1:** Update the nominal $\lambda_{UE2}^{*}=\Upsilon_{UE2}^{\left(i\right)}$ by (62).3: **Step 2.1:** Update $\beta_{UE2}^{o\left(i\right)}$ by (65).4: **Step 2.2:** Update the transmission beamforming $\alpha_{2}^{o\left(i\right)}$ by (66).5: **Step 3:** Update $\Theta^{\left(i\right)}$ by (71).Until the value of the function $ASR_{UE2}\left(\alpha_{2},\Theta ,\lambda_{UE2}\right)$ in (60) converges.

## 5. Simulation Results and Discussion

In this section, we simulate the analyzed results using the Monte Carlo simulation method in MATLAB software, version R2014a. The simulation settings we set are as shown in Table 1.

In Figure 4, we simulate the SOPs of the UEs according to the SNR, in which the IRS has 10, 20, and 30 reflectors with NOMA power allocation factors of 0.3 and 0.7 for UE1 and UE2, respectively. The simulation analysis results of the UEs’ SOPs show us that all the simulation and analysis results are consistent. When the number of IRS reflectors is large, the slope of the SOP characteristics is large and is achieved quickly when the SNR is small. Therefore, we can confirm that the security performance of the system depends on the number of IRS reflectors. In addition, we see that the SOP characteristics of the UEs also depend on the NOMA power allocation factor of the UE. Specifically, if the NOMA power allocation factor of UE2 is larger than UE1, the characteristic SOP curve of UE2 has a lower SNR value than the SOP of UE1. To evaluate the influence of the NOMA power allocation factor, we continue to simulate the SOPs of UEs with different NOMA power allocation factors as shown in Figure 5.

In Figure 5, we simulate the SOP of UE users according to the SNR, where the IRS has 20 reflective elements with NOMA power allocation factors of 0.2, 0.4, and 0.7 for UE1 and NOMA power allocation factors of 0.8, 0.6, and 0.3 for UE2, respectively. The simulation results for the UE SOPs match the analysis results for the UE SOPs. From the analysis results in Figure 4 and Figure 5, it can be seen that the SOPs depend not only on the number of IRS elements but also on the NOMA power allocation factor for each UE. Thus, with the DIRS-NOMA system model, we propose that if the system is designed with a large number of IRS reflective elements plus a reasonable NOMA power allocation factor, the system’s security performance will be high.

In Figure 6, we simulate the SOP of the DIRS-NOMA system and the NOMA system, where the IRS has 20 reflectors with NOMA power allocation coefficients of 0.6 for UE1 and 0.4 for UE2, respectively. The simulation results for the SOPs of the UEs match the analysis results for the SOPs of the UEs. The curves representing the SOP of the DIRS-NOMA system have a much higher slope than the curves representing the SOP of the NOMA system. This proves that the DIRS-NOMA system has the ability to improve security performance much better than the NOMA system. Specifically, when the SNR reaches 10 dB, the SOP of the DIRS-NOMA system reaches about $10^{-4}$, while the SOP of the NOMA system reaches below $10^{-1}$.

In Figure 7, we use simulations to optimize the ASRs of the UEs according to the SNR, where the IRS has 10 and 20 reflectors and the power allocation factors are 0.1 and 0.9 for UE1 and UE2, respectively. The results of the analysis of the ASRs of the UEs are consistent with the results of the simulation of the ASRs of the corresponding UEs. From the simulation analysis results, we can see that the ASR characteristics reach their maximum values at different SNR values. This proves that the ASR characteristics of the UEs depend on the number of reflectors of the IRS. In addition, the maximum point of the ASR characteristics of the UEs has a small maximum value when the NOMA power allocation factor is small and reaches a large value when the NOMA power allocation factor for the UE is large. To evaluate the influence of the NOMA power allocation factor on the ASR of UEs, we continue to simulate the ASRs of UEs with different NOMA power allocation factors, as shown in Figure 8.

In Figure 8, we simulate the ASRs of the UEs according to the SNR, where the IRS has 10 reflectors. The NOMA power allocation factors are 0.2, 0.4, and 0.7 for UE1 and 0.8, 0.6, and 0.3 for UE2, respectively. The simulation results show us that when the power allocation factors change, the characteristic curve of the ASR changes. The maximum value of ASR achieved by different UEs is different and depends on the NOMA power allocation factors for the UEs. In addition, according to the characteristic curve, we see that the maximum ASR values of the UEs are achieved at a constant SNR and do not depend on the NOMA power allocation factors.

From the above analysis, we can see that the security performance of the proposed DIRS-NOMA system depends not only on the NOMA power allocation factor but also on the number of IRS reflection elements. If we design the number of IRS elements and the NOMA power allocation factor reasonably, we can improve the security performance of the system. We can confirm that an IRS has the ability to improve the security performance of the DIRS-NOMA system.

## 6. Conclusions

In this paper, we computed the security performance parameters of a DIRS-NOMA system supporting communication with a group of two NOMA UEs at the edge, without direct connection to the BS, where there is an ED associated with each UE. We calculated and derived the closed-form expressions of the SOP and ASR for the UEs of the DIRS-NOMA system. From the closed-form expression of the ASR, we proved that this parameter reaches its maximum value at a given SNR. The maximum value of the ASR depends only on the number of IRS reflection elements and does not depend on the NOMA power allocation factor. We have proven and verified the theoretical arguments through simulation. Through analysis and discussion of the results above, we have confirmed that if we design the number of IRS elements and the NOMA power allocation factor reasonably, we can improve the security performance of the system, and an IRS is capable of improving the security performance of the DIRS-NOMA system. In addition, we formulate and solve an ASR optimization problem by optimizing the NOMA power allocation factor combined with IRS reflection beam optimization by the method to maximize WSR using the fractional programming technique. Furthermore, through the analysis, simulation, and discussion of the results above, it has been demonstrated that the DIRS-NOMA technique is capable of improving the security performance of the system. We hope that the analysis and discussion of the security performance of the DIRS-NOMA system model that we propose will provide useful contributions to other studies in the future.

## Figures and Tables

**Figure 1 sensors-25-01274-f001:**
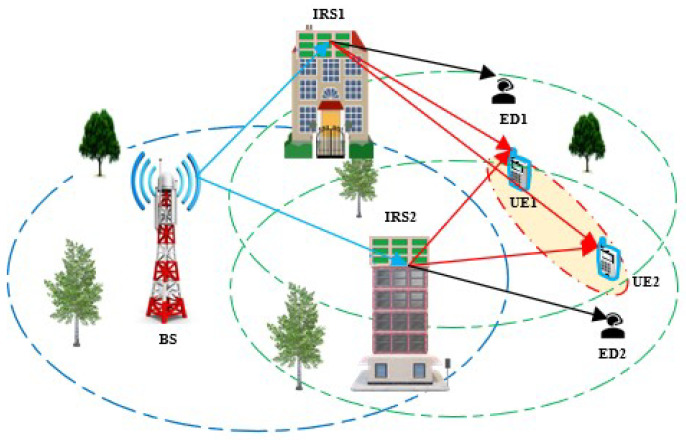
DIRS-NOMA network model.

**Figure 2 sensors-25-01274-f002:**
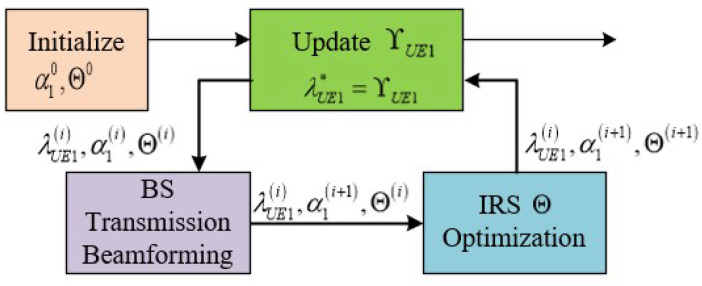
Alternating optimization for (42a)–(42c).

**Figure 3 sensors-25-01274-f003:**
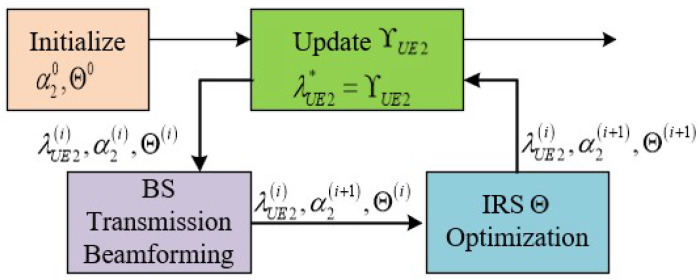
Alternating optimization for (59a)–(59c).

**Figure 4 sensors-25-01274-f004:**
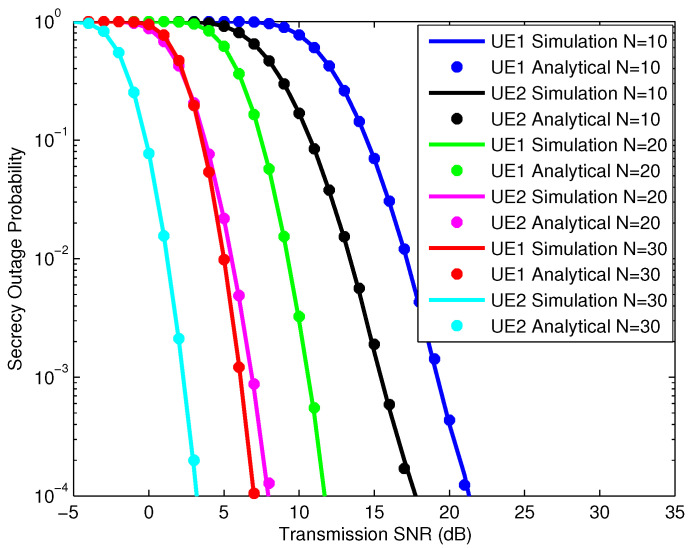
SOP of UEs with different numbers of IRS reflection elements.

**Figure 5 sensors-25-01274-f005:**
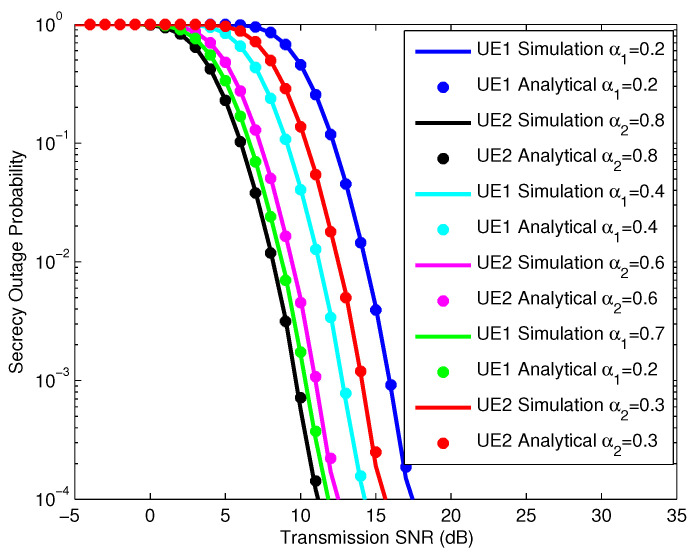
SOP of UEs with different power allocation factors.

**Figure 6 sensors-25-01274-f006:**
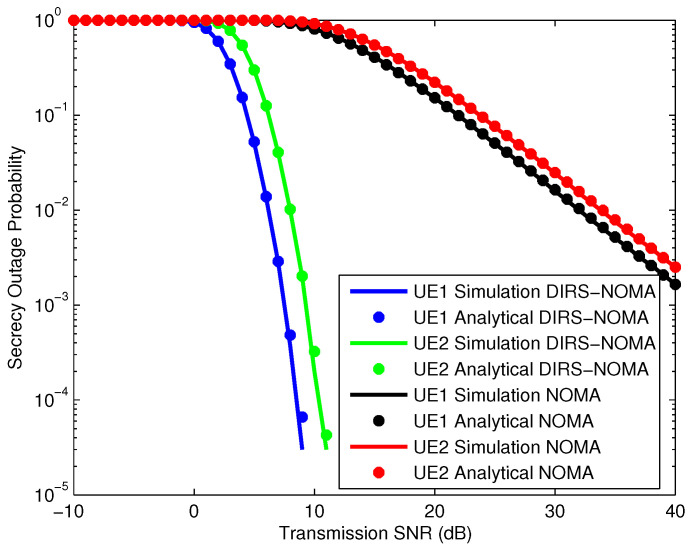
Comparison of SOPs between DIRS-NOMA and NOMA systems.

**Figure 7 sensors-25-01274-f007:**
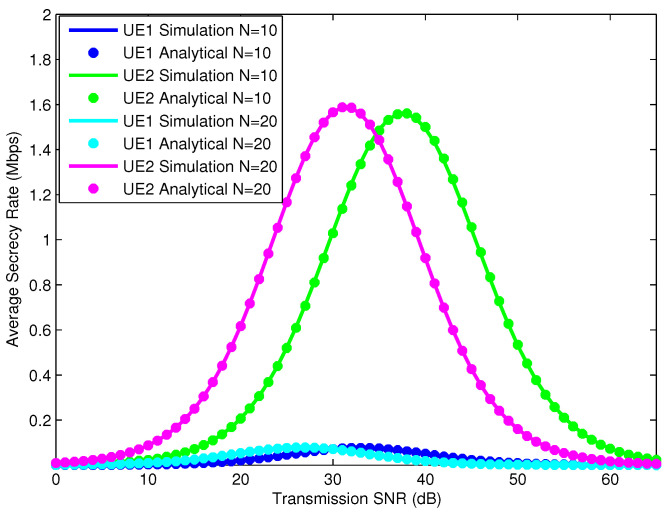
ASR of UEs with different numbers of IRS reflection elements.

**Figure 8 sensors-25-01274-f008:**
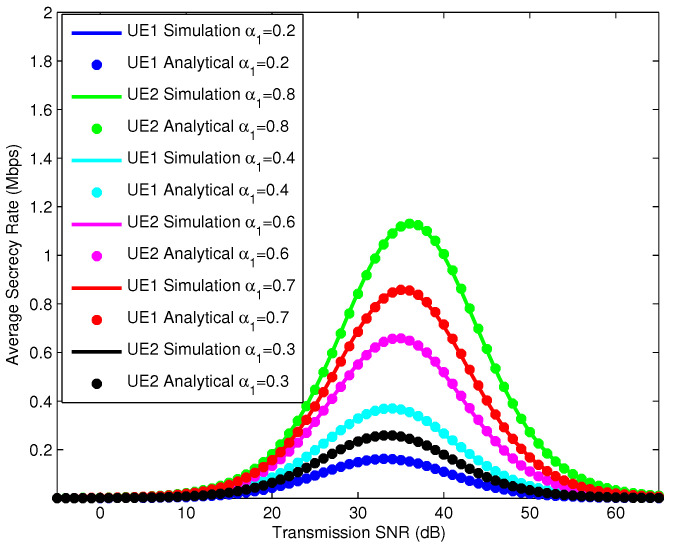
ASR of UEs with different NOMA power allocation factors.

**Table 1 sensors-25-01274-t001:** Simulation parameters.

No.	Meaning	Parameters
1	Bandwidth	B = 1 Mhz
2	Amplitude-reflection coefficient	β=0.9
3	Distance $d_{\mathrm{BR}1}$	20 m
4	Distance $d_{\mathrm{BR}2}$	20 m
5	Distance $d_{11}$	10 m
6	Distance $d_{22}$	10 m
7	Distance $d_{12}$	8 m
8	Distance $d_{21}$	8 m
9	Distance $d_{ED1}$	13 m
10	Distance $d_{ED2}$	13 m
11	Path-loss exponent $\alpha_{G}$	2.5
12	Path-loss exponent $\alpha_{g}$	2.5
13	Target data rates for fixed-rate transmission	$C_{th}=0.01$

## Data Availability

Data are contained within the article.

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
