# Peer review of "Security Performance Analysis of Downlink Double Intelligent Reflecting Surface Non-Orthogonal Multiple Access Network for Edge Users"

_sensors, 2025, doi:10.3390/s25041274_

Round 1

Reviewer 1 Report

Comments and Suggestions for Authors

  1. The author has studied the security performance of the intelligent reflecting surface-enabled communication networks. While the benefit of the IRS technology has been discussed, the necessity of introducing NOMA technology is not evident, and make the research appear to be a mere assemblage of concepts.
  2. If the introduction of NOMA technology is indeed necessary, the authors should clarify the scenarios in which such a problem urgently needs to be addressed. Moreover, the effectiveness and necessity of applying NOMA technology should be thoroughly discussed. It would be much better to analyze the advantages of NOMA in comparison with other spectrum efficiency enhancement technologies, such as cognitive radio and integrated sensing and communication technologies, by discussing the following references.

[1] "Integrating Sensing and Communications for Ubiquitous IoT: Applications, Trends, and Challenges," IEEE Network, vol. 35, no. 5, pp. 158-167, 2021.
[2] "Cooperative Trajectory Planning and Resource Allocation for UAV-Enabled Integrated Sensing and Communication Systems," IEEE Transactions on Vehicular Technology, vol. 73, no. 5, pp. 6502-6516, 2024.

[3] "Advances in cognitive radio networks: A survey," IEEE Journal of Selected Topics in Signal Processing, vol. 5, no. 1, pp. 5-23, 2011.

  1. The literature review lacks logicality and relevance to the authors' research content (Physical layer security). Many recent literature reviews are omitted, such as: [1]"Robust Design of the Security Scheme in IRS-Assisted MISO Systems With Imperfect Eavesdropping CSI," IEEE Transactions on Vehicular Technology, vol. 73, no. 9, pp. 12815-12827, 2024.
    [2]"IRS-Assisted Physical Layer Security in MIMO-NOMA Networks," IEEE Communications Letters, vol. 27, no. 3, pp. 792-796, 2023.
    [3]"Robust and Secure Sum-Rate Maximization for Multiuser MISO Downlink Systems With Self-Sustainable IRS," IEEE Transactions on Communications, vol. 69, no. 10, pp. 7032-7049, 2021.
  2. The new objective function in (43) should not be identical to that in (41a). The authors should use two distinct variables to represent them.
  3. The convergence of the proposed algorithm should be analyzed and discussed.
  4. There still exists several typo in the references. The authors should more carefully proofread the paper.
Comments on the Quality of English Language

Overall, the English language used in the manuscript requires significant improvement.

Author Response

We sincerely thank the reviewer for his interesting and meaningful comments that helped us improve our article. We have explained your comments in the attached PDF file. We hope you are satisfied with our response. We wish you good health and success in your work.

Reviewer 2 Report

Comments and Suggestions for Authors

In signal analysis section, the authors are saying BS has 1 antenna and then they are saying it has M antennas? This makes paper extremely difficult to understand.

2. A lot of typos are there in the paper. Authors at some places have also mentioned Rayleigh as Reyleigh.

3. The work is not very novel. Assumption of perfect CSI and SIC simplifies the problem too much. 

4. What is the AWGN value?

5. Comparison with benchmark scheme hasn't been included which limits the contribution of this work.

6. 

Author Response

(The authors gave the same response as above.)

Reviewer 3 Report

Comments and Suggestions for Authors

In this paper, the security performance of the DIRS-NOMA wireless communication system supporting communication with a group of two NOMA users (UE) at the edge, without direct connection to the base station (BS), with existence of eavesdropping user (ED) is studied. The security outage probability (SOP) and the average security rate (ASR) at each UE user is derived. The main idea is interesting, however, following comments need to be addressed to improve the quality of the paper.

1). In the abstract provide numerical values for the performance gains of the proposed IRS assisted NOMA compared to the existing methods.

2). Some abbreviations have not been defined in the main text. Please check carefully and correct (e.g., DPSFA, DPDAV, SOP, ASR, NOMA, RIS, ..., etc). Also make sure abbreviations are consistent (e.g., IRS, RIS, ..., etc).

3). The contributions of the paper should be well organized. You can remove the detailed description on the system model from the first contribution and present in the previous sentence. Better to add some interesting numerical findings in the final contribution point. 

4). Fig. 1 should be more clearer. Put a legend to describe critical details such as user groups etc.

5). "According to the decoding principle of NOMA, UEs directly decode their own signals by treating signals intended for other users as interference.". Is this valid for all NOMA users? To the best of reviewer's knowledge, the near user need to decode the far user's signal first. Otherwise there is a practical issue of decoding the symbols as SINR is not at the required level for certain \alpha_1 and \apha_2 values. This should be clearly described since it is critical for the accuracy. Else explain the actual decoding principle. The reviewer recommends following papers to refer.

a). Z. Ding, X. Lei, G. K. Karagiannidis, R. Schober, J. Yuan and V. K. Bhargava, "A Survey on Non-Orthogonal Multiple Access for 5G Networks: Research Challenges and Future Trends," in IEEE Journal on Selected Areas in Communications, vol. 35, no. 10, pp. 2181-2195, Oct. 2017.

b). K. W. S. Palitharathna, H. A. Suraweera, R. I. Godaliyadda, V. R. Herath and J. S. Thompson, "Average Rate Analysis of Cooperative NOMA Aided Underwater Optical Wireless Systems," in IEEE Open Journal of the Communications Society, vol. 2, pp. 2292-2310, 2021.

6). The parameters of each expression should be presented in the paragraph after the term "where", since are also in the same sentence.

7). In the optimization problem in 37, provide optimization parameters.

8). The algorithm formats should be improved.

9). The results of the paper should be well improved. Most obvious results are presented. Present with different system parameters.

10). Can you compare the simulation results with existing models or conventional NOMA?

11). The reference format of the paper is poor. Use a consistent format. Also for journals, use abbreviations and for conferences use "in Proc. xxx".

Author Response

(The authors gave the same response as above.)

Round 2

Reviewer 1 Report

Comments and Suggestions for Authors

The authors have addressed the issues I raised previously. However, regarding the reorganization of the introduction, the authors have improved the necessity of using the IRS technology and the research status of physical layer security. Nevertheless, the necessity of NOMA technology was not explained, that is, its superiority over the CR (UAV-Assisted Secure Communications in Terrestrial Cognitive Radio Networks: Joint Power Control and 3D Trajectory Optimization) and ISAC (Cooperative Trajectory Planning and Resource Allocation for UAV-Enabled Integrated Sensing and Communication Systems) technologies. Also, it is required to analyze the existing bottlenecks for NOMA technology to highlight the contributions of this paper.

Author Response

We sincerely thank you very much for your important comments that help us improve our article. We have improved our article in the attached file. We hope that this improvement will satisfy you.

Reviewer 2 Report

Comments and Suggestions for Authors

I would like to thank the authors for making the changes as per our comments. However, I would like to highlight that my comments 2 and 3 (Reviewer 2) have not been addressed properly. I would highly recommend to at least include impact of imperfect SIC to make the study more realistic.

Additionally, what is noise power. In many papers people take value in between -65dbm to -110 dbm. Authors can then convert it to linear scale. What value authors have considered for simulations.

Author Response

(The authors gave the same response as above.)

Reviewer 3 Report

Comments and Suggestions for Authors

The authors have well address all the comments. No further comments from the reviewer.

Author Response

We sincerely thank you very much for your important comments that help us improve our article. 
